# Alternative Hapten Design for Zearalenone Immunoreagent Generation

**DOI:** 10.3390/toxins14030185

**Published:** 2022-03-02

**Authors:** Antonio Abad-Fuentes, Consuelo Agulló, Daniel López-Puertollano, Ismael Navarro-Fuertes, Antonio Abad-Somovilla, Josep Vicent Mercader

**Affiliations:** 1Instituto de Agroquímica y Tecnología de Alimentos, Spanish Council for Scientific Research (CSIC), 28006 Madrid, Spain; aabad@iata.csic.es; 2Department of Organic Chemistry, University of Valencia, 46010 València, Spain; consuelo.agullo@uv.es (C.A.); daniel.lopez@uv.es (D.L.-P.); isnafue@uv.es (I.N.-F.); antonio.abad@uv.es (A.A.-S.)

**Keywords:** mycotoxin, hapten design, spacer arm, antibody affinity, antibody specificity, immunoassay

## Abstract

Appropriate hapten design and synthesis have been identified as critical steps to generate high-performance immunoreagents and to develop sensitive and selective immunoanalytical methods. Antibodies and immunoassays for the major mycotoxin zearalenone have been reported and marketed. However, zearalenone haptens have mostly been prepared by the oxime active ester technique, and hapten characterization has generally been poor or non-existent. In the present study, novel haptens of zearalenone with longer linkers and with alternative tethering sites have been designed for immunizing and assay conjugate preparation. All of these molecules were purified and spectroscopically verified, and a structure-activity relationship evaluation was carried out. This approach revealed that the hapten with the linker at the carbonyl group generated antibodies with a higher affinity than the hapten functionalized at the phenyl moiety. Antibodies produced with the latter hapten, on the other hand, showed lower cross-reactivity values to the major zearalenone metabolites. Finally, similar immunoassay sensitivity was achieved with all of the antibodies when heterologous haptens were employed. Furthermore, by altering the structure of the competing antigen, the immunoassay selectivity was modified. These results demonstrate that immunochemical methods for zearalenone rapid analysis can still be improved in terms of sensitivity and selectivity.

## 1. Introduction

Mycotoxins are the group of chemical contaminants potentially present in food and feed for which immunochemical techniques enjoy a higher degree of implementation and acceptability in analytical laboratories around the world, even being supported and recommended by regulatory authorities in certain cases. Immunoanalytical methods are based on the high-affinity, selective, reversible, and non-covalent binding between a target substance (analyte) and an antibody. Depending on the analytical requirements, antibodies can be integrated into a variety of immunoanalytical methods, including immunoaffinity columns, lateral-flow immunochromatography assays (immunostrips), biosensors, microarrays, and particularly the enzyme-linked immunosorbent assay (ELISA) [1].

In order to generate antibodies to low molecular weight compounds like mycotoxins, the target molecule must be covalently coupled to an immunogenic macromolecule, such as a protein. To prepare such a bioconjugate, however, the target compound must be chemically modified by introducing a functionalized aliphatic linker at specific positions of the molecular framework. The functional group is commonly located at the end of the spacer arm, and it should be easily activated for protein coupling under aqueous conditions. The structural and electronic similarity between the synthetic functionalized analogue, so-called hapten, and the target molecule, together with the manner in which the skeleton of the target molecule is exposed to the immune system, have a significant impact on the binding properties of the subsequently generated antibody. Therefore, hapten design and synthesis are widely regarded as critical steps in the development of antibody-based techniques for the detection of chemical contaminants. In particular, the position of the spacer arm on the target molecule has been revealed to be of the utmost importance for the generation of antibodies with the desired affinity and specificity [2]. Nevertheless, predicting the optimum linker tethering site is frequently difficult. In our experience, molecular modelling computational tools may provide relevant clues for optimal hapten design, even though the synthesis of different regioisomeric haptens with well-defined anchoring sites is still the most productive approach [3].

Zearalenone (ZEN), a macrocyclic β-resorcyclic acid lactone (Figure 1), is a mycotoxin produced by *Fusarium* species. It is classified as an estrogenic compound because of its structural similarity to naturally occurring estrogens, and it induces obvious adverse effects in humans and other animals [4]. ZEN is one of the most prevalent mycotoxins worldwide, and it is frequently found in corn, wheat, barley, and other cereal grains, and occasionally in milk, meat, and eggs [5,6,7]. As a result, the European Union has set maximum permitted limits (MPL) for this mycotoxin in grain, ranging from 75 to 400 μg/kg depending on the commodity and the intended use. For processed cereal-based foods intended for consumption by infants and young children, a more restrictive limit of 20 μg/kg has been established [8,9]. Currently, a number of ZEN metabolites have been identified (Figure 1), mainly zearalanone (ZAN), α-zearalenol (α-ZEL), β-zearalenol (β-ZEL), α-zearalanol (α-ZAL), and β-zearalanol (β-ZAL), albeit they are not included in the EU legislation. α-ZAL is used as a growth promoter in some countries under the name of zeranol; however, this compound is banned in the EU, hence it is included in many official control programs [10].

The approaches so far reported for synthesizing ZEN haptens have mostly involved the use of a simple ZEN-oxime derivative at position C-7 [11,12]. In our opinion, this chemical strategy suggests relevant alterations in the stereoelectronic properties and conformation of the parent molecule. The aim of the present study was to investigate the structure-activity relationship of immunogens, both in terms of affinity and specificity of the generated antibodies. With this goal in mind, we have designed two novel ZEN regioisomeric haptens in which a 5-carbon atom functionalized spacer arm is incorporated at opposite sites of the mycotoxin molecular skeleton. Competitive ELISA in two different formats was used to investigate the antibodies’ binding properties. Moreover, additional haptens with structural modifications were prepared in order to evaluate the influence of hapten heterology on immunoassay sensitivity and selectivity.

## 2. Results and Discussion

### 2.1. Design and Synthesis of Immunizing Haptens

The covalent attachment of the whole ZEN skeleton to the carrier protein by the oxime active ester method, via prior conversion of the carbonyl group at C-7 into the corresponding *O*-(carboxymethyl)oxime (CMO) derivative, has been mostly used to prepare the immunizing bioconjugate [13,14,15]. Other approaches, such as the glutaraldehyde method [16,17], the Mannich-type condensation reaction [18], and the crosslinking reaction with 1,4-butanediol diglycidyl ether [19], have been employed on a few occasions. Except for the oxime active ester method, where C-7 carbonyl group uniquely determines the conjugation position, none of the other conjugation procedures allows precise control of the linker tethering site and, in any case, none of the ZEN derivatives prepared by these procedures have been isolated and structurally characterized prior to conjugation. As a result, no clear relationship can be established between the hapten’s derivatization position and the affinity and specificity of the generated antibodies. On the other hand, most of the anti-ZEN antibodies produced so far following these approaches showed significant cross-reactivity with ZEN metabolites [20,21], most likely because of a questionable choice of the functionalization site.

In this study, we have designed two new immunizing haptens of ZEN. One of them, hapten ZE*o*, incorporates a ((5-carboxypentyl)oxy)imino spacer arm through the carbonyl function at C-7, whereas hapten ZE*p* incorporates a carboxylated five-carbon long aliphatic spacer arm through the oxygenated quasi-distal position at C-14 (Appendix A). In principle, the longer arm of hapten ZE*o* should enable for better exposure to the immune system of the entire ZEN skeleton than in the case of the analogous hapten with a CMO spacer arm prepared by most authors. As a result, the presentation of the most functionalized and, in theory, most antigenic region of ZEN is optimized. On the other hand, the incorporation of the spacer arm through the C-14 position, as in hapten ZE*p*, maximizes the exposure of the macrocyclic lactone ring without causing significant alterations in its conformational arrangement relative to ZEN itself. It must be taken into account that the ZEN molecule has a great conformational flexibility associated with the rotation of the C–C bonds that are part of the 14-member macrocyclic system, particularly around the C-3 to C-11 bonds. Nevertheless, all theoretical and experimental studies indicate that ZEN exists in a preferred conformation, both in solid state and in aqueous solution, in which the methyl and carbonyl groups at C-3 and C-7, respectively, are oriented in the same direction towards the outside of the molecule. However, in solution, another slightly less stable conformation (0.8 kcal/mol) may exist, in which both groups are oriented in opposite directions [22]. Both conformations are stabilized by an intramolecular hydrogen bond between the hydroxyl group at C-16 and the oxygen atom of the lactonic carbonyl group. The position and mode of attachment of the ZEN skeleton to the carrier protein may play an important role in the immune response, not only because of its electronic and steric influence, but also because it may condition the spatial conformational arrangement of the macrolactonic ring.

These two new haptens were prepared from ZEN using efficient synthetic procedures, as outlined in Figure 2. Hapten ZE*o* was prepared in a single step by reacting ZEN in pyridine with 6-(aminooxy)hexanoic acid hydrochloride (**1**), which was made from 6-bromohexanoic acid in four steps: *tert*-butyl ester protection, nucleophilic substitution of the bromine atom by the *N*-hydroxyphthalimidyl group, hydrazinolysis to remove the phthalimidyl amino protecting group, and final cleavage of the *tert*-butyl ester with trifluoroacetic acid [23]. Hapten ZE*o*, which was isolated in high purity and with excellent yield without further purification after a simple workup, was obtained as a ca. 3:2 mixture of geometric isomers around the C=N double bond. The presence of the two geometric isomers, *E* and *Z*, is clearly evident in the ^1^H and ^13^C NMR spectra, where almost all signals are doubled (see the Appendix A). The significant differences in the chemical shifts of the protons observed for some of the signals of the two isomers, such as those of the H-11′ olefinic protons (5.98 vs. 5.75 ppm) or the H-3′ methine protons (5.15 vs. 4.99 ppm), as well as the different pattern of the associated coupling constants, suggest a dependence of the conformation of the macrocyclic lactone ring on the geometry of the oximino moiety. In particular, the chemical shifts and the H-H coupling pattern of the major C-7 oxime isomer clearly correlate with those reported in the literature for ZEN in the same solvent (methanol-d_4_) [24], suggesting that this isomer exhibits similar conformational behavior in solution such as ZEN.

On the other hand, two steps were required to produce hapten ZE*p* from ZEN. First, an *O*-alkylation reaction with *tert*-butyl 5-bromovalerate was carried out under standard Williamson ether synthesis conditions (**2**) [25]. This alkylation reaction yielded a ca. 3:1 mixture of di- and mono-*O*-alkylation products, which were easily separated by column chromatography to provide the product resulting from the selective *O*-alkylation of the more reactive C-14 hydroxyl group, i.e., **3**, in 60% yield. The synthesis of hapten ZE*p* was readily completed from intermediate **3** by cleavage of the *tert*-butyl ester group to give the corresponding carboxylic moiety. This was done by brief exposure of **3** to trifluoroacetic acid in CH_2_Cl_2_ to *yield hapten* ZE*p*
*in virtually quantitative yield.* Hapten ZE*p* has very similar spectroscopic properties compared to ZEN—except for the minor changes caused by the linker at C-14—suggesting that this hapten and the toxin have analogous electronic and conformational characteristics, as expected (see the Appendix A).

### 2.2. Heterologous Haptens

Two haptens (ZE*oh* and ZE*ph*) with structural modifications compared to the immunizing haptens were also designed (Appendix A). Hapten ZE*oh* is related to the immunizing hapten ZE*o* since they have the same linker tethering site, but the heterologous hapten contains a methoxy group at C-14 instead of the hydroxyl group, whereas hapten ZE*ph* holds the spacer arm at the same position as ZE*p* and differs from it by the functionalization of the C-16 position.

The synthesis of the heterologous hapten ZE*oh* was accomplished from known 14-*O*-methylzearalenone following a procedure similar to that used for the preparation of hapten ZE*o* (Figure 2). This new hapten was also obtained as a ca. 3:2 mixture of *E* and *Z* geometric isomeric oximes. The synthesis of the heterologous hapten ZE*ph* was carried out from the intermediate of the synthesis of hapten ZE*p*, the *tert*-butyl ester **3**, in just two steps (Figure 2). First, methylation of the phenolic hydroxyl group at C-16 by treatment of **3** with dimethyl sulfate and catalytic K_2_CO_3_, followed by cleavage of the *tert*-butyl ester group with trifluoroacetic acid. Overall, both reactions proceed efficiently to afford hapten ZE*ph* in an overall yield of 86%.

### 2.3. Hapten Activation

After completing the synthesis of all haptens, their carboxylic group was activated via their transformation into the corresponding *N*-hydroxysuccinimidyl ester, a necessary step in the preparation of the immunizing and assay conjugates. This reaction was carried out under standard activation conditions in *N*,*N*-dimethylformamide (DMF) at room temperature using carbodiimide (EDC∙HCl) and *N*-hydroxysuccinimide (NHS), giving the corresponding *N*-hydroxysuccinimidyl esters, ZE*o*-NHS, ZE*oh*-NHS, ZE*p*-NHS, and ZE*ph*-NHS (Figure 2), in good yield. Because of the relative lability of the *N*-hydroxysuccinimidyl ester moiety, these active esters were structurally characterized by ^1^H NMR after their preparation to check purity and integrity, and they were employed immediately for protein coupling.

### 2.4. Bioconjugate Preparation

Haptens with defined linker tethering sites were employed to synthesize the immunizing and assay protein conjugates. Moreover, these bioconjugates were prepared using essentially pure NHS esters of the haptens to better control the coupling reaction, thus reducing the amount of required hapten and the formation of undesired byproducts. The immunizing conjugates with bovine serum albumin (BSA) were prepared to achieve high hapten-to-protein molar ratios, i.e., 17.6 and 15.6 for haptens ZE*o* and ZE*p*, respectively (Appendix A). In contrast, low hapten densities were targeted for the ovalbumin (OVA) conjugates in order to get optimal immunoassay sensitivity. Therefore, the molar ratios for the four conjugates ranged from 1.5 to 3.5. Finally, the molar ratios of the enzyme tracers were lower (between 0.5 and 1.5) due to the limited availability of Lys residues in horseradish peroxidase (HRP). BSA and OVA conjugates were stored frozen at −20 °C in PB, whereas the enzyme tracers were kept at 4 °C in phosphate buffered saline (PBS) containing 1% BSA (*w*/*v*) and 0.01% (*w*/*v*) thimerosal. However, the enzyme tracers of haptens ZE*o* and ZE*oh*—those containing two or one unprotected OH groups at a distal position, respectively—lost their antigenic activity within a few weeks, most likely due to functional modifications of the conjugated hapten molecule, as the enzyme activity was not affected. In contrast, the HRP conjugates with haptens ZE*p* and ZE*ph*—in which at least one of the two OH groups are protected and the linker is located at a proximal site—remained antigenically active in the long term. New batches of the ZE*o* and ZE*oh* enzyme tracers were prepared and, on this occasion, the bioconjugates were diluted 1:3 with ethylene glycol and stored at −20 °C.

### 2.5. Antibody Generation and Characterization

Two antibodies were obtained using the BSA conjugates of ZEN haptens with opposite linker tethering sites, i.e., haptens ZE*o* and ZE*p*. These new immunoreagents were evaluated by direct antibody-coated and indirect conjugate-coated competitive ELISA (d-cELISA and i-cELISA, respectively) using the homologous antigen (the conjugate carrying the same hapten as the immunizing conjugate). Under these conditions, the affinity of the antibodies that were derived from hapten ZE*o* was higher than those of the antibodies obtained from hapten ZE*p* (Table 1), probably due to the different antigenic properties of the most exposed regions in each hapten, that is, the polysubstituted aromatic system of hapten ZE*o* versus the very flexible aliphatic macrocyclic ring of hapten ZE*p*. The affinity of these antibodies was equivalent to that of the best previously reported polyclonal antibodies to ZEN [17,26].

Additionally, antibody specificity was assessed by d-cELISA and i-cELISA using the homologous antigen. Cross-reactivity (CR) was determined for ZAN, α-ZEL, β-ZEL, α-ZAL, and β-ZAL, using ZEN as reference. Similar results were observed with the two cELISA formats. ZE*o*-derived antibodies showed high or moderate binding to all metabolites, except for β-ZAL (Figure 3). A similar CR pattern was observed by Usleber et al. using an equivalent immunizing hapten for polyclonal antibody generation [26]. On the contrary, ZE*p*-derived antibodies were more specific for ZEN, and only ZAN was generally bound. Antibody ZE*p*#2 did not recognize the metabolites with a reduced carbonyl group at C-7 (Figure 1 and Figure 2). Therefore, hapten ZE*p* generated more specific antibodies than hapten ZE*o*. These results demonstrate the major influence of the linker tethering site on the affinity and specificity of ZEN antibodies.

### 2.6. Evaluation of Heterologous Antigens

Two types of heterologous antigens (conjugates carrying a hapten different from that of the immunizing conjugate) were evaluated. First, antigens whose only heterology was the position of the linker, that is, antigens with hapten ZE*p* for ZE*o*-derived antibodies and with hapten ZE*o* for ZE*p*-derived antibodies, were assayed. The binding of the four antibodies was examined in both immunoassay formats using the corresponding antigens. No binding between antibodies and these heterologous combinations was observed in the d-cELISA format (Appendix A). Concerning the indirect assays, ZE*o*-type antibodies did not recognize the OVA conjugate of ZE*p*, but ZE*p*-type antibodies did bind the conjugate of ZE*o*. Interestingly, the sensitivity of the i-cELISA was improved by approximately one order of magnitude when the antigen OVA–ZE*o* in combination with ZE*p*-type antibodies were used, resulting in the best IC_50_ values in this assay format (Figure 4).

On the other hand, antigens with two heterologies were also assayed by direct and indirect cELISA. Haptens ZE*oh* and ZE*ph* were double heterologous haptens of haptens ZE*p* and ZE*o*, respectively (Appendix A). ZE*o*-derived antibodies bound ZE*oh* antigens; however, the IC_50_ values increased by d-cELISA and stayed almost unchanged by i-cELISA (Table 2) compared with the assays using the homologous antigen (Table 1). As expected, bioconjugates with hapten ZE*ph* were not bound by ZE*o*-type antibodies in any of the assayed cELISA formats. Concerning ZE*p*-derived antibodies, binding to the heterologous antigen of hapten ZE*ph* was observed in both immunoassay formats (Table 2), and low IC_50_ values were obtained. Interestingly, the enzyme tracer of hapten ZE*oh* was also recognized by these antibodies, and very low IC_50_ values were found. Finally, the IC_50_ values obtained with antigens OVA-ZE*o* and OVA-ZE*oh* were comparable.

It is well known that the selectivity of competitive immunoassays strongly depends on the antigen’s structure when polyclonal antibodies are used [27,28,29]. For this reason, CR values for the major ZEN metabolites were verified using heterologous bioconjugates (Figure 3 and Appendix A). When OVA-ZE*oh* was used for microplate coating, the immunoassay with ZE*o*-derived antibodies increased the CR values to α-ZEL. On the other hand, the metabolite α-ZEL was generally not recognized by ZE*p*-type antibodies, but the CR was around 50% for antibody ZE*p*#1 when OVA-ZE*o* and OVA-ZE*oh* were employed as coating antigens. Therefore, heterologous bioconjugates helped not only to improve immunoassay sensitivity, but also to modulate selectivity.

## 3. Conclusions

A series of novel zearalenone haptens have been synthesized and spectroscopically characterized. Moreover, active NHS esters were employed for protein coupling. Thus, bioconjugates with unambiguous chemical compositions were prepared in order to evaluate the influence of the linker tethering site in ZEN haptens over the immune response. Antibodies to zearalenone that were raised using hapten ZE*o*—with the spacer arm distal to the aromatic moiety of the mycotoxin—showed higher affinity than those obtained from hapten ZE*p*, displaying the aliphatic macrocyclic ring of the molecule. Interestingly, when heterologous haptens with opposite linker tethering sites were used, immunoassay sensitivity with ZE*p*-type antibodies was similar to that of the immunoassays obtained with ZE*o*-type antibodies. On the other hand, higher specificity was achieved with the immunogen of the novel hapten ZE*p*. In fact, antibody ZE*p*#2 showed cross-binding only with the synthetic metabolite ZAN. Our results pave the way for obtaining high-affinity monoclonal antibodies and developing immunoassays to ZEN with new binding features by using these previously unreported haptens.

## 4. Materials and Methods

### 4.1. Reagents and Instruments

All solvents were purified by distillation and, if necessary, they were dried using standard methods [30]. Air- and moisture-sensitive reactions were carried out under a positive pressure of nitrogen using material previously oven-dried at 130 °C overnight. Reactions were monitored using thin-layer chromatography with 0.25 mm pre-coated silica gel plates. The plates were visualized under UV light at 366 and 254 nm, using an aqueous ceric ammonium molybdate solution or an ethanolic phosphomolybdic acid solution and heat as developing agents. Chromatography refers to flash column chromatography, which was performed on silica gel 60 (particle size 40−63 μm) with the given solvents. ^1^H/^13^C NMR spectra were recorded at 298 °K, in the indicated solvent, at 300/75 MHz (Bruker Avance DPX300, Billerica, MA, USA) or 500/126 MHz (Bruker Avance DRX500). ^1^H and ^13^C chemical shifts (δ scale) are expressed in parts per million (ppm) downfield from tetramethylsilane and are referenced to residual proton or carbon in the NMR solvent (CHCl_3_: δ 7.26/77.16; methanol-d_4_: δ 3.31/40.00). A combination of COSY, edited HSQC, and HMBC experiments was used to assign the ^1^H and ^13^C chemical shifts of selected compounds. Accurate mass measurements (HRMS) were obtained using the electrospray ionization (ESI) mode on a Q-TOF premier mass spectrometer with an electrospray source from Waters (Manchester, UK).

Standard ZEN ((4*S*,12*E*)-16,18-dihydroxy-4-methyl-3-oxabicyclo(12.4.0)octadeca-1(14),12,15,17-tetraene-2,8-dione, CAS registry number 17924-92-4, Mw 318.4) was purchased from Fermentek (Jerusalem, Israel). Metabolites (ZAN, α-ZEL, β-ZEL, α-ZAL, and β-ZAL) were obtained also from Fermentek. Standard solutions were prepared in anhydrous DMF, and the stock solutions were stored at −20 °C. PBS 10× solution (Fisher BioReagents BP399-20) was from Thermo Fisher Scientific (Waltham, MA, USA). Fraction V BSA, from Roche Applied Science (Mannheim, Germany) was employed to synthesize the immunizing bioconjugates. OVA, HRP, complete and incomplete Freund’s adjuvants, adult bovine serum (ABS), and *o*-phenylenediamine (OPD) were acquired from Merck (Darmstadt, Germany). Sephadex G-25 HiTrap^®^ Desalting columns for protein–hapten conjugate purification were obtained from GE Healthcare (Uppsala, Sweden) and operated under an ÄKTA Purifier workstation also from GE Healthcare. A 5800 matrix-assisted laser desorption ionization time-of-flight (MALDI-TOF/TOF) mass spectrometry apparatus from ABSciex (Framingham, MA, USA) was used for bioconjugate analysis. Immunoassays were carried out using Costar^®^ 96-well flat-bottom high-binding polystyrene ELISA plates from Corning (Corning, NY, USA). Polyclonal goat anti-rabbit immunoglobulins antibody (GAR) for microplate coating and GAR conjugated to HRP (GAR-HRP) for competitive ELISA analysis were purchased from Rockland Immunochemicals Inc. (Pottstown, PA, USA) and BioRad (Madrid, Spain), respectively. Microplate wells were washed with an ELx405 washer and immunoassay absorbance values were read with a PowerWave HT microplate reader, both from BioTek Instruments (Winooski, VT, USA).

### 4.2. Solutions and Buffers

Coating buffer: 50 mM carbonate-bicarbonate buffer, pH 9.6; enzyme substrate solution: 2 mg/mL of OPD in 25 mM citrate and 62 mM phosphate buffer, pH 5.4, containing 0.012% (*v*/*v*) H_2_O_2_; HEPES buffer: 20 mM HEPES, pH 7.4; PB: 100 mM phosphate buffer, pH 7.4; PBS: 12 mM phosphate containing 137 mM NaCl and 2.7 mM KCl, pH 7.4; PBST: PBS containing 0.05% (*v*/*v*) Tween-20; secondary antibody solution: 10^4^-fold diluted GAR-HRP in PBST containing 10% (*v*/*v*) ABS; washing solution: 150 mM NaCl containing 0.05% (*v*/*v*) Tween-20. All buffers and solutions were prepared in Milli-Q^®^ water.

### 4.3. Synthesis of Immunizing Haptens

Preparation of 6-((((*S*,*E*)-14,16-dihydroxy-3-methyl-1-oxo-1,3,4,5,6,8,9,10-octahydro-7H-benzo[c][1]oxacyclotetradecin-7-ylidene)amino)oxy)hexanoic acid (hapten ZE*o*). A solution of ZEN (12.8 mg, 40.2 µmol) and 6-(aminooxy)hexanoic acid hydrochloride [23] (**1**, 10 mg, 54.5 µmol) in anhydrous pyridine (370 µL) was stirred in the dark at room temperature for 24 h under nitrogen. The pyridine was removed with a stream of nitrogen and the resulting residue was dissolved in EtOAc, washed with water and brine, dried under anhydrous Na_2_SO_4_, and concentrated under a vacuum to give hapten ZE*o* (17.5 mg, 97%)—a mixture of two geometrical isomers (*E* and *Z*) of the oximino moiety—as a viscous colorless oil. ^1^H NMR (300 MHz, methanol-d_4_) δ 6.98 and 6.96 (each br d, *J* = 15.5 Hz, 1H, H-12′), 6.39 and 6.36 (each d, *J* = 2.5 Hz, 1H, H-13′), 6.22 and 6.21 (d, *J* = 2.6 Hz, 1H, H-15′), 5.98 (dt, *J* = 15.5, 6.4 Hz, 0.4H, H-11′), 5.75 (ddd, *J* = 15.4, 9.4, 4.6 Hz, 0.6H, H-11′), 5.15 and 4.99 (each m, 1H, H-3′), 4.03 and 4.00 (each t, *J* = 6.3 Hz, 2H, H_2_-6), 2.84 (m, 0.6H, H-8′), 2.53 (dt, *J* = 13.9, 7.2 Hz, 0.4H, H-6′), 2.40–2.00 (m, 7H), 1.98–1.40 (m, 12H), 1.38 and 1.36 (each d, *J* = 6.2 Hz, 3H, Me-3′); ^13^C NMR (75 MHz, methanol-d_4_), δ 177.6 (C-1), 172.4 and 172.4 (C-1′), 166.0 and 164.7 (C-14′), 163.7 and 163.3 (C-16′), 162.1 and 159.5 (C-7′), 145.0 and 143.7 (C-12′a), 133.9 and 132.9 (C-12′), 133.3 and 133.0 (C-11′), 109.4 and 108.5 (C-13′), 106.2 and 104.4 (C-16′a), 102.7 (C-15′), 74.3 and 73.9 (C-3′), 74.1 and 74.0 (C-6), 36.8 and 36.2 (C-4′), 35.6 and 31.2 (C-8′), 35.0 (C-2), 31.9 and 31.8 (C-10′), 30.1 and 29.9 (C-5), 28.4 and 27.5 (C-6′), 27.0 and 26.8 (C-4), 26.0 and 25.9 (C-3), 25.7 and 23.8 (C-9′), 23.3 and 23.2 (C-5′), 20.9 and 20.5 (Me-3′); HRMS (ESI) *m*/*z* calculated for C_24_H_34_NO_7_ [M + H]^+^ 448.2330, found 448.2324.

Preparation of *tert*-butyl (*S*,*E*)-5-((16-hydroxy-3-methyl-1,7-dioxo-3,4,5,6,7,8,9,10-octahydro-1H-benzo[c][1]oxacyclotetradecin-14-yl)oxy)pentanoate (**3**). A heterogeneous mixture of ZEN (32.7 mg, 0.103 mmol), *tert*-butyl 5-bromovalerate [25] (**2**, 36.5 mg, 154 mmol), tetrabutylammonium iodide 3.6 mg, 9.5 µmol), and K_2_CO_3_ (21.9 mg, 0.158 mmol) in anhydrous acetone (2.2 mL) was stirred at 55 °C overnight under nitrogen. The reaction mixture was cooled to room temperature, diluted with Et_2_O (50 mL) and washed with water and brine, dried over anhydrous MgSO_4,_ and concentrated at reduced pressure. The obtained residue (70 mg) was purified by column chromatography, using hexane/EtOAc 9:1 as eluent, to give the 14-alkylate derivative **3** (29.3 mg, 60%) as an oil, followed by the 14,16-dialkylated product (13.5 mg, 21%). ^1^H NMR (300 MHz, CDCl_3_) δ 12.06 (s, 1H, 16′-OH), 7.01 (dd, *J* = 15.3, 1.9 Hz, 1H, H-12′), 6.45 (dd, *J* = 2.7, 0.5 Hz, 1H, H-13′), 6.36 (d, *J* = 2.6 Hz, 1H, H-15′), 5.68 (ddd, *J* = 15.2, 10.5, 3.7 Hz, 1H, H-11′), 5.00 (m, 1H, H-3′), 3.98 (d, *J* = 5.8 Hz, 2H, H_2_-5), 2.84 (ddd, *J* = 18.8, 12.2, 2.7 Hz, 1H, H-8′), 2.60 (dt, *J* = 12.6, 4.5 Hz, 1H, H-6′), 2.40–2.08 (m, 5H), 2.29 (t, *J* = 6.9 Hz, 2H, H_2_-2), 1.86–1.57 (m, 9H), 1.45 (s, 9H, CMe_3_), 1.38 (d, *J* = 6.1 Hz, 3H, Me-3′); ^13^C NMR (75 MHz, CDCl_3_) δ 211.2 (C-7′), 172.9 (C-1), 171.5 (C-1′), 165.7 (C-16′), 163.6 (C-14′), 143.4 (C-12′a), 133.4 (C-11′), 132.5 (C-12′), 108.7 (C-15′), 103.6 (C-16′a), 100.5 (C-13′), 80.4 (CMe_3_), 73.5 (C-3′), 67.7 (C-5), 44.1 (C-6′), 36.8 (C-8′), 35.2 (C-2), 34.9 (C-4′), 31.1 (C-10′), 28.5 (C-4), 28.3 (CMe_3_), 22.4, 21.8, and 21.2 (C-3, C-5′, and C-9′), 21.0 (Me-3′); HRMS (ESI) *m*/*z* calculated for C_27_H_39_O_7_ [M + H]^+^ 475.2690, found 475.2670.

Preparation of (*S*,*E*)-5-((16-hydroxy-3-methyl-1,7-dioxo-3,4,5,6,7,8,9,10-octahydro-1H-benzo[c][1]oxacyclotetradecin-14-yl)oxy)pentanoic acid (hapten ZE*p*). Trifluoroacetic acid (0.8 mL) was dropwise added to a solution of *tert*-butyl ester **3** (13.5 mg, 28.4 μmol) in anhydrous CH_2_Cl_2_ (0.8 mL) cooled to 0 °C under nitrogen and the resultant solution was stirred for 1 h. The solvents were evaporated under vacuum and the remaining traces of trifluoroacetic acid were removed by repetitive evaporations with ethanol-free chloroform to afford hapten ZE*p* (11.6 mg, 98%) as an amorphous solid. ^1^H NMR (300 MHz, CDCl_3_), δ 12.09 (s, 1H, 16′-OH), 7.37 (br s, 1H, OH), 7.01 (dd, *J* = 15.3, 2.0 Hz, 1H, H-12′), 6.45 (d, *J* = 2.5 Hz, 1H, H-13′), 6.37 (d, *J* = 2.6 Hz, 1H, H-15′), 5.68 (ddd, *J* = 14.8, 10.4, 3.6 Hz, 1H, H-11′), 5.10–4.90 (m, 1H, H-3′), 3.99 (d, *J* = 5.4 Hz, 2H, H_2_-5), 2.86 (ddd, *J* = 18.5, 12.2, 2.6 Hz, 1H, H-8′), 2.68–2.55 (m, 1H, H-6′), 2.53–2.29 (m, 3H), 2.26–2.97 (m, 4H), 1.90–1.42 (m, 9H), 1.38 (d, *J* = 6.1 Hz, 3H, Me-3′); ^13^C NMR (75 MHz, CDCl_3_) δ 211.9 (C-7′), 171.5 (C-1′), 165.7 (C-16′), 163.6 (C-14′), 143.5 (C-12′a), 133.4 (C-11′), 132.5 (C-12′), 108.7 (C-15′), 103.7 (C-16′a), 100.5 (C-13′), 73.5 (C-3′), 67.6 (C-5), 43.1 (C-6′), 36.8 (C-8′), 34.9 (C-4′), 33.6 (C-2), 31.1 (C-10′), 28.4 (C-4), 22.4, 21.4, and 21.1 (C-3, C-5′, and C-9′), 21.0 (Me-3′); HRMS (ESI) *m*/*z* calculated for C_23_H_31_O_7_ [M + H]^+^ 419.2064, found 419.2073.

### 4.4. Synthesis of Heterologous Haptens

Preparation of 6-((((*S*,*E*)-16-hydroxy-14-methoxy-3-methyl-1-oxo-1,3,4,5,6,8,9,10-octahydro-7H-benzo[c][1]oxacyclotetradecin-7-ylidene)amino)oxy)hexanoic acid (hapten ZE*oh*). A solution of 14-*O*-methylzearalenone [31] (**5**, 7.5 mg, 22.6 μmol) and hydrochloride **1** (5.0 mg, 27.3 μmol) in anhydrous pyridine (200 μL) was stirred at room temperature for 24 h under nitrogen. After removing the pyridine with a stream of nitrogen, the residue was dissolved in EtOAc and successively washed with water, 0.1 M aqueous HCl, 5% aqueous NaHCO_3_, and brine. Drying of the organic phase over anhydrous MgSO_4_ and evaporation of the solvent under reduced pressure gave hapten ZE*oh* (10.2 mg, 98%)—a mixture of *E* and *Z* oximes – as a colorless oil. ^1^H NMR (500 MHz, methanol-d_4_) δ 6.96 and 6.93 (each br d, *J* = 15.5 Hz, 1H, H-12′), 6.49 and 6.46 (each m, 1H, H-13′), 6.36 (m, 1H, H-15′), 6.04 and 5.79 (each m, 1H, H-11′), 5.16 and 5.01 (each m, 1H, H-3′), 4.03 and 4.00 (each t, *J* = 6.4 Hz, 2H, H_2_-6), 3.80 (s, 3H, OMe), 2.82 (m, 0.6H, H-8′), 2.49 (dt, *J* = 14.3, 7.2 Hz, 0.4Hz, H-8′), 2.40–2.02 (m, 7H), 1.95–1.52 (m, 10H), 1.50–1.40 (m, 2H), 1.39 and 1.37 (each d, *J* = 6.4 Hz, 3H, Me-3′); ^13^C NMR (126 MHz, methanol-d_4_), δ 177.7 (C-1), 172.6 and 172.2 (C-1′), 165.9 and 165.3 (C-14′), 164.8 and 164.1 (C-16′), 162.2 and 159.5 (C-7′), 145.5 and 143.0 (C-12′a), 133.7 and 132.4 (C-12′), 133.6 and 133.5 (C-11′), 108.5 and 107.2 (C-13′), 107.9 and 105.5 (C-16′a), 100.9 and 100.8 (C-15′), 74.5 and 73.9 (C-3′), 74.1 and 74.0 (C-6), 55.9 (OMe), 36.8 and 36.2 (C-4′), 35.5 and 31.3 (C-8′), 35.1 (C-2), 31.9 and 31.8 (C-10′), 30.1 and 29.9 (C-5), 28.5 and 27.5 (C-6′), 27.0 and 26.8 (C-4), 26.0 (C-3), 25.6 and 23.8 (C-9′), 23.3 (C-5′), 20.9 and 20.5 (Me-3′); HRMS (ESI) *m*/*z* calculated for C_25_H_36_NO_7_ [M + H]^+^ 462.2486, found 462.2473.

Preparation of *tert*-butyl (*S*,*E*)-5-((16-methoxy-3-methyl-1,7-dioxo-3,4,5,6,7,8,9,10-octahydro-1H-benzo[c][1]oxacyclotetradecin-14-yl)oxy)pentanoate (**4**). A solution of Me_2_SO_4_ (6.6 mg, 4.9 μL, 52 μmol) in dry acetone (26 μL) was added to a mixture of *tert*-butyl ester **3** (13.8 mg, 29.1 μmol), and K_2_CO_3_ (6.0 mg, 43.4 μmol) in acetone (0.45 mL) at room temperature under nitrogen. The resulting mixture was stirred under the same conditions for 48 h, then diluted with water and extracted with EtOAc. The combined organic layers were washed with water and brine and dried under anhydrous Na_2_SO_4_. The residue that was left after evaporation of the solvent at reduced pressure was chromatographed on silica gel, using a 99:1 CHCl_3_/MeOH mixture as eluent, to give compound **4** (13.5 mg, 95%) as a colorless oil. ^1^H NMR (500 MHz, CDCl_3_) δ 6.58 (d, *J* = 2.1 Hz, 1H, H-13′), 6.38 (dd, *J* = 15.5, 1.5 Hz, 1H, H-12′), 6.36 (d, *J* = 2.0 Hz, 1H, H-15′), 5.99 (ddd, *J* = 15.5, 10.0, 4.3 Hz, 1H, H-11′), 5.30 (ddd, *J* = 10.0, 6.4, 3.3 Hz, 1H, H-3′), 3.99 (t, *J* = 6.0 Hz, 2H, H_2_-5), 3.80 (s, 3H, OMe), 2.70 (ddd, *J* = 16.7, 11.2, 3.3 Hz, 1H, H-8′), 2.45-2.24 (m, 3H), 2.30 (t, *J* = 7.3 Hz, 2H, H_2_-2), 2.20–1.97 (m, 3H), 1.90–1.50 (m, 9H), 1.45 (s, 9H, CMe_3_), 1.34 (d, *J* = 6.3 Hz, 3H, Me-3′); ^13^C NMR (126 MHz, CDCl_3_) δ 211.6 (C-7′), 172.9 (C-1), 167.8 (C-1′), 160.9 (C-16′), 157.8 (C-14′), 136.9 (C-12′a), 133.3 (C-11′), 129.2 (C-12′), 116.4 (C-16′a), 102.1 (C-13′), 98.3 (C-15′), 80.4 (CMe_3_), 71.4 (C-3′), 67.8 (C-5), 56.1 (OMe), 44.3 (C-6′), 37.8 (C-8′), 35.3 (C-4′), 35.2 (C-2), 31.4 (C-10′), 28.7 (C-4), 28.3 (CMe_3_), 22.0, 21.8, and 21.5 (C-3, C-5′, and C-9′), 20.2 (Me-3′); HRMS (ESI) *m*/*z* calculated for C_28_H_41_O_7_ [M + H]^+^ 489.2847, found 489.2857.

Preparation of (*S*,*E*)-5-((16-methoxy-3-methyl-1,7-dioxo-3,4,5,6,7,8,9,10-octahydro-1H-benzo[c][1]oxacyclotetradecin-14-yl)oxy)pentanoic acid (hapten ZE*ph*). This hapten was prepared as described above for hapten ZE*p*, using *tert*-butyl ester **4** (9.2 mg, 18.8 mmol), anhydrous CH_2_Cl_2_ (0.48 mL), and trifluoroacetic acid (0.50 mL). After workup of the reaction mixture, the obtained crude product was purified by chromatography on silica gel, using CHCl_3_/MeOH mixtures (from 99:1 to 95:5) as eluent, to afford hapten ZE*ph* (7.4 mg, 91%) as an amorphous white solid. ^1^H NMR (300 MHz, CDCl_3_) δ 6.58 (d, *J* = 2.0 Hz, 1H, H-13′), 6.38 (dd, *J* = 15.5, 1.5 Hz, 1H, H-12′), 6.36 (d, *J* = 2.0 Hz, 1H, H-15′), 5.99 (ddd, *J* = 15.6, 9.9, 4.3 Hz, 1H, H-11′), 5.31 (m, 1H, H-3′), 4.01 (t, *J* = 7.0 Hz, 2H, H_2_-5), 3.80 (s, 3H, OMe), 2.70 (ddd, *J* = 17.6, 11.6, 3.4 Hz, 1H, H-8′), 2.49–2.25 (m, 5H), 2.20–1.97 (m, 3H), 1.89–1.50 (m, 9H), 1.34 (d, *J* = 6.2 Hz, 3H, Me-3′); ^13^C NMR (126 MHz, CDCl_3_) δ 211.8 (C-7′), 178.4 (C-1), 167.8 (C-1′), 160.8 (C-16′), 157.8 (C-14′), 137.0 (C-12′a), 133.3 (C-11′), 129.2 (C-12′), 116.4 (C-16′a), 102.1 (C-13′), 98.3 (C-15′), 71.4 (C-3′), 67.7 (C-4), 56.1 (OMe), 44.2 (C-6′), 37.8 (C-8′), 35.3 (C-4′), 33.5 (C-2), 31.4 (C-10′), 28.6 (C-4), 22.0 (C-5′), 21.5 (C-3 and C-9′), 20.2 (Me-3′); HRMS (ESI) *m*/*z* calculated for C_24_H_33_O_7_ [M + H]^+^ 433.2221, found 433.2217.

### 4.5. Hapten Activation

Preparation of the *N*-hydroxysuccinimidyl ester of hapten ZE*o* (ZE*o*-NHS ester). A solution of hapten ZE*o* (15.4 mg, 34.4 µmol), *N*-(3-dimethylaminopropyl)-*N*′-ethylcarbodiimide hydrochloride (7.8 mg, 40.7 µmol, 1.2 equiv), and *N*-hydroxisuccinimide (6.0 mg, 52.1 µmol, 1.5 equiv) in anhydrous DMF (0.5 mL) was stirred at room temperature overnight under nitrogen. The reaction mixture was diluted with Et_2_O and successively washed with water, 5% aqueous NaHCO_3_, 1.5% aqueous LiCl, and brine. The organic layer was dried over anhydrous MgSO_4_ and concentrated under reduced pressure to give the *N*-hydroxysuccinimidyl ester of hapten ZE*o**,* ZE*o*-NHS ester (15.5 mg, 92% of crude product)—a mixture of *E* and *Z* oximes—as a colorless oil, which was used immediately for the preparation of the corresponding protein and enzyme conjugates. ^1^H NMR (300 MHz, CDCl_3_) δ 12.00 and 11.91 (each s, 1H, 16′-OH), 7.07 and 7.00 (each br d, *J* = 15.6 Hz, 1H, H-12′), 6.40 and 6.39 (each d, *J* = 2.6 Hz, 1H, H-13′), 6.33 and 6.32 (d, *J* = 2.6 Hz, 1H, H-15′), 5.94 (dt, *J* = 15.7, 6.6 Hz, 0.4H, H-11′), 5.74 (ddd, *J* = 15.4, 8.7, 5.4 Hz, 0.6H, H-11′), 5.16 and 5.01 (each m, 1H, H-3′), 4.04 and 4.03 (each t, *J* = 6.4 Hz, 2H, H_2_-6), 2.83 (s, 4H, COCH_2_CH_2_CO), 2.63 and 2.62 (each t, *J* = 7.5 Hz, 2H, H_2_-2), 2.40–2.00 (m, 6H), 1.96–1.43 (m, 12H), 1.39 and 1.38 (each d, *J* = 6.2 Hz, 3H, Me-3′).

Preparation of the *N*-hydroxysuccinimidyl ester of hapten ZE*p* (ZE*p*-NHS ester). ZE*p*-NHS ester was prepared as described above for ZE*o*-NHS, using hapten ZE*p* (11.3 mg, 27 µmol), EDC·HCl (7.3 mg, 38.1 µmol, 1.4 equiv), NHS (4.8 mg, 41.7 µmol, 1.5 equiv), and DMF (0.5 mL). Crude ZE*p*-NHS ester (11.9 mg, 85%) was obtained as a foam. ^1^H NMR (300 MHz, CDCl_3_) δ 12.06 (s, 1H, 16′-OH), 7.01 (dd, *J* = 15.3, 2.0 Hz, 1H, H-12′), 6.45 (d, *J* = 2.5 Hz, 1H, H-13′), 6.37 (d, *J* = 2.5 Hz, 1H, H-15′), 5.69 (ddd, *J* = 15.3, 10.4, 3.7 Hz, 1H, H-11′), 5.05–4.94 (m, 1H, H-3′), 4.01 (d, *J* = 5.5 Hz, 2H, H_2_-5), 2.84 (s, 4H, COCH_2_CH_2_CO), 2.78–2.90 (m overlapped with signal at 2.84, 1H, H-8′), 2.70 (t, *J* = 7.0 Hz, 2H, H_2_-2), 2.64–2.55 (m, 1H, H-6′), 2.47–2.30 (m, 2H), 2.30–1.44 (m, 12H), 1.38 (d, *J* = 6.1 Hz, 3H, Me-3′).

Preparation of the *N*-hydroxysuccinimidyl ester of hapten ZE*oh* (ZE*oh*-NHS ester). ZE*oh*-NHS ester was prepared as described above for ZE*o*-NHS, using hapten ZE*oh* (9.8 mg, 21.2 µmol), EDC·HCl (5.0 mg, 26.1 µmol, 1.2 equiv), NHS (3.3 mg, 28.7 µmol, 1.4 equiv), and DMF (0.35 mL). Crude ZE*oh*-NHS ester (11.2 mg, 95%)—a mixture of *E* and *Z* oximes—was obtained as an oil. ^1^H NMR (500 MHz, CDCl_3_) δ 12.03 and 11.93 (each s, 1H, 16′-OH), 7.08 and 7.02 (dd, *J* = 15.7, 1.7 Hz, 1H, H-12′), 6.46 and 6.45 (each d, *J* = 2.6 Hz, 1H, H-13′), 6.39 and 6.38 (each d, *J* = 2.6 Hz, 1H, H-15′), 5.95 (ddd, *J* = 15.6, 7.4, 5.7 Hz, 0.4H, H-11′), 5.75 (ddd, *J* = 15.5, 9.5, 4.6 Hz, 0.6H, H-11′), 5.17 and 5.00 (each m, 1H, H-3′), 4.05 and 4.02 (each t, *J* = 6.4 Hz, 2H, H_2_-6), 3.82 and 3.81 (each s, 3H, OMe), 2.82 (br s, 4H, COCH_2_CH_2_CO), 2.63 and 2.62 (each t, *J* = 7.5 Hz, 2H, H_2_-2), 2.40–2.04 (m, 6H), 1.96–1.45 (m, 12H), 1.40 and 1.39 (each d, *J* = 6.3 Hz, 3H, Me-3′).

Preparation of the *N*-hydroxysuccinimidyl ester of hapten *ZEph* (ZE*ph*-NHS ester). ZE*ph*-NHS ester was prepared as described above for ZE*o*-NHS, using hapten ZE*ph* (7.7 mg, 17.8 µmol), EDC·HCl (4.0 mg, 20.9 µmol, 1.2 equiv), NHS (3.0 mg, 26.1 µmol, 1.5 equiv), and DMF (0.25 mL). Crude ZE*ph*-NHS ester (7.8 mg, 83%) was obtained as a foam. ^1^H NMR (500 MHz, CDCl_3_) δ 6.59 (d, *J* = 2.1 Hz, 1H, H-13′), 6.37 (dd, *J* = 15.6, 1.8 Hz, 1H, H-12′), 6.36 (d, *J* = 2.0 Hz, 1H, H-15′), 6.00 (ddd, *J* = 15.5, 10.0, 4.3 Hz, 1H, H-11′), 5.30 (ddt, *J* = 10.0, 6.3, 3.3 Hz, H-3′), 4.03 (t, *J* = 5.8 Hz, 2H, H_2_-5), 3.80 (s, 3H, OMe), 2.84 (s, 4H, COCH_2_CH_2_CO), 2.71 (t, *J* = 6.9 Hz, 2H, H_2_-2), 2.70 (ddd, *J* = 17.0, 11.1, 3.2 Hz, 1H, H-8′), 2.41 (ddd, *J* = 13.6, 9.4, 4.6 Hz, 1H, H-6′), 2.36–2.26 (m, 2H), 2.19–1.51 (m, 12H), 1.33 (d, *J* = 6.3 Hz, 3H, Me-3′).

### 4.6. Bioconjugate Preparation and Analysis

Protein conjugates were prepared by the active ester method. The purified hapten-NHS esters were dissolved in DMF to obtain 50 mM solutions. The four haptens were coupled to OVA and HRP, whereas only haptens ZE*o* and ZE*p* were conjugated to BSA. Protein solutions were prepared in PB at 15 mg/mL for BSA and OVA and at 3 mg/mL for HRP. Conjugation reactions were carried out overnight at room temperature in amber glass vials by drop wise adding the hapten-NHS solution over the protein solution under vigorous stirring. To prepare BSA conjugates, a 30-fold hapten-to-protein molar ratio was used whereas molar excess values around 12 were employed to prepare OVA and HRP conjugates. The conjugates were purified by size-exclusion chromatography using PB as eluent, with the exception of tracers HRP-ZE*o* and HRP-ZE*oh* that were purified with HEPES buffer. Fractions containing the bioconjugates were pooled and processed as described in previous studies [32].

Analysis of hapten bioconjugates was performed by MALDI-TOF/TOF mass spectrometry as follows. Briefly, 100 μL of protein or enzyme conjugate solutions (*ca*. 1 mg/mL and 0.5 mg/mL for BSA/OVA and HRP conjugates, respectively) were dialyzed against Milli-Q^®^ water. The final volume of each dialysis solution was ca. 200 μL. Then, 0.8 μL of every sample solution was spotted onto the MALDI plate. After, the droplets were air-dried at room temperature, and 0.8 μL of matrix (10 mg/mL of sinapinic acid (Bruker) in 70% MeCN, 0.1% TFA) was added and allowed to air-dry at room temperature. The resulting mixtures were analyzed by MALDI-TOF/TOF in positive linear mode (1500 shots every position) in a mass range of 12,000-100,000 *m*/*z*, with laser intensity of 6000. Previously, the plate was calibrated with 1 μL of the TOF/TOF calibration mixture (ABSciex), in 13 positions. Every sample was calibrated by a ‘close external calibration’ method with a BSA, OVA, or HRP spectrum acquired in a close position. The analysis of the results was performed using the mMass program (v5.5.0, available at http://www.mmass.org/, accessed on 3 February 2022).

### 4.7. Antibody Generation

Animal manipulation was carried out according to European Directive 2010/63EU and Spanish laws (RD118/2021 and law 32/2007) regarding the protection of experimental animals. A group of two female New Zealand white rabbits were immunized by subcutaneous injection of each BSA conjugate. Animals received, every three weeks, 300 μg of protein conjugate in a 1:1 water-in-oil emulsion (1 mL) between PBS and Freund’s adjuvant. Complete adjuvant was employed for the first injection, and it was substituted by incomplete adjuvant for subsequent boosts. Ten days after the fourth immunization, animals were exsanguinated, and the blood was left overnight at 4 °C for coagulation. The cells were separated from the serum by centrifugation during 20 min at 3000× *g*. The antisera were partially purified by salting out twice with one volume of a cold saturated (3.9 M) solution of ammonium sulfate. Antibodies were preserved as precipitates at 4 °C. For daily usage, a fraction of antiserum was diluted in PBS containing 1% BSA and 0.01% thimerosal.

### 4.8. Competitive ELISA

Plate microwells were washed four times with washing solution after each incubation step. For antibody-coated direct competitive assays, plates were coated by overnight incubation at 4 °C with 100 μL per well of GAR solution (1 μg/mL) in coating buffer. Then, 100 μL per well of antiserum dilution in PBST was added, and plates were incubated 1 h at room temperature. The competitive reaction was carried out at room temperature during 1 h by mixing 50 μL per well of standard solution in PBS and 50 μL per well of enzyme tracer solution in PBST. The retained enzyme activity was revealed during 10 min at room temperature by adding 100 μL per well of freshly prepared enzyme substrate solution. The chromogenic reaction was stopped with 100 μL per well of 1 M H_2_SO_4_.

For the conjugate-coated indirect competitive ELISA format, microplates were coated by overnight incubation at room temperature with 100 μL per well of bioconjugate solution in coating buffer. The competitive reaction was performed with 50 μL per well of standard solution in PBS and 50 μL per well of antibody dilution in PBST, and incubation at room temperature for 1 h. Next, 100 μL per well of secondary antibody solution was applied, and the plates were incubated again 1 h at room temperature. Finally, color was developed as for the previous format.

The absorbance of the chromogenic reaction products in the microwells was read at 492 nm with 650 nm as reference wavelength. Seven calibration solutions of each analyte were prepared in borosilicate glass vials by serial dilution in PBS from the most concentrated solution. A blank solution was assayed together with the standards. Inhibition curves were obtained by fitting the experimental values to a four-parameter logistic equation for standard curves using the SigmaPlot software from Systat Software Inc. (v14.5, San Jose, CA, USA, 2020). The A_max_ value corresponds to the maximum absorbance value obtained in the absence of analyte. The antibody apparent affinity was estimated as the half-maximum inhibition concentration (IC_50_) of the analyte. Cross-reactivity (%) was calculated from the quotient between the IC_50_ for ZEN and the IC_50_ for the assayed metabolite.

## Figures and Tables

**Figure 1 toxins-14-00185-f001:**
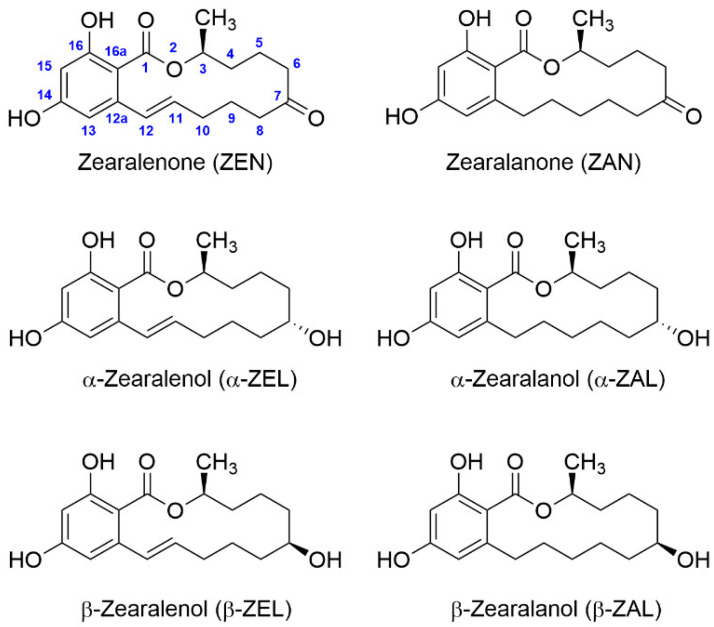
Chemical structure of zearalenone and its major metabolites.

**Figure 2 toxins-14-00185-f002:**
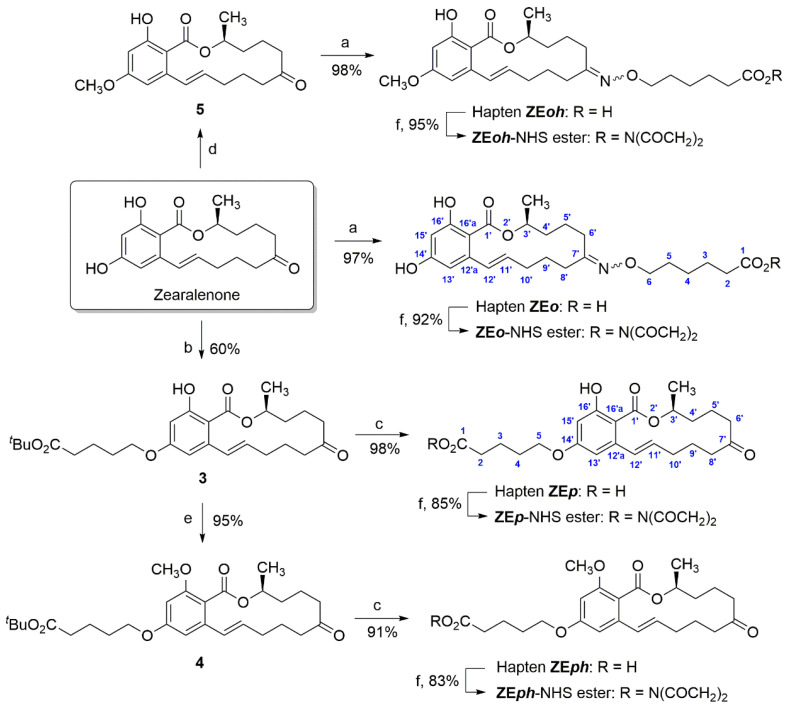
Synthesis of haptens and the corresponding *N*-hydroxysuccinimidyl ester. Reagents and conditions: (a) HO_2_C(CH_2_)_5_ONH_2_∙HCl (1), pyridine, room temperature, 24 h; (b) Br(CH_2_)_4_CO_2_*^t^*Bu (2), K_2_CO_3_, Bu_4_NI, acetone, 55 °C, overnight; (c) CF_3_CO_2_H, CH_2_Cl_2_, 0 °C to room temperature, 1 h; (d) K_2_CO_3_, CH_3_I, acetone, room temperature, overnight; (e) K_2_CO_3_, (CH_3_)_2_SO_4_, acetone, room temperature, 48 h; (f) EDC·HCl, NHS, DMF, room temperature, overnight.

**Figure 3 toxins-14-00185-f003:**
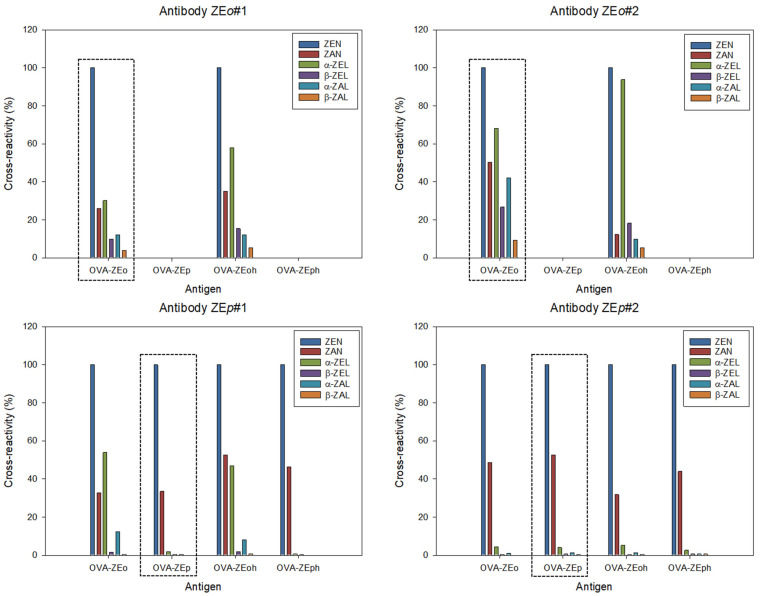
Immunoassay selectivity using different competitive assay antigens determined by i-cELISA. The results obtained using the corresponding homologous bioconjugate are framed.

**Figure 4 toxins-14-00185-f004:**
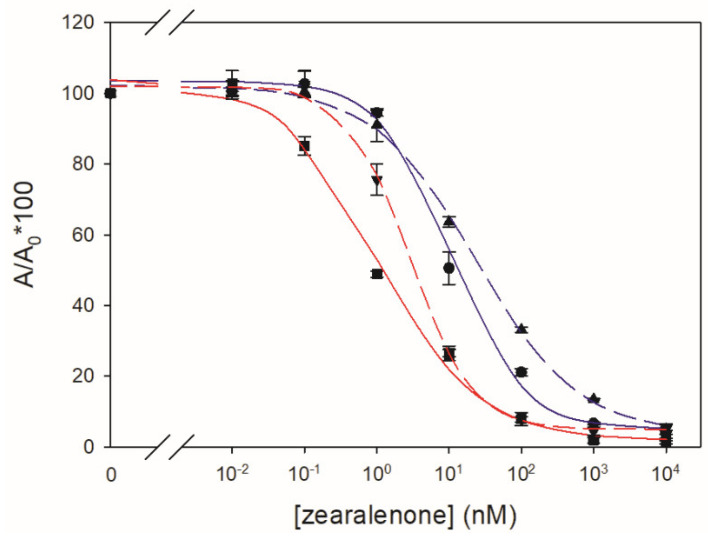
Inhibition curves (*n* = 3) obtained by i-cELISA using antibodies ZE*p*#1 (solid lines) and ZE*p*#2 (dashed lines) with homologous (blue) and heterologous (red) conjugates.

**Table 1 toxins-14-00185-t001:** Antibody characterization by competitive ELISA using the homologous bioconjugates (*n* = 3) ^a^.

Ab	d-cELISA			i-cELISA		
	[Ab] ^b^	[HRP] ^c^	IC_50_ ^d^	[Ab]	[OVA]	IC_50_
ZE*o*#1	5	100	3.1	45	100	1.5
ZE*o*#2	5	10	7.1	45	100	5.2
ZE*p*#1	5	10	11.5	15	10	10.8
ZE*p*#2	5	30	14.1	45	100	23.4

^a^ A_max_ values were between 0.5 and 1.5. ^b^ Antibody dilution factor ×10^−3^. ^c^ Bioconjugate concentrations are in ng/mL. ^d^ Values are in nM units.

**Table 2 toxins-14-00185-t002:** Antibody characterization using double heterologous bioconjugates (*n* = 3) ^a^.

	d-cELISA						i-cELISA					
	HRP-ZE*oh*			HRP-ZE*ph*			OVA-ZE*oh*			OVA-ZE*ph*		
Ab	[Ab] ^b^	[OVA] ^c^	IC_50_ ^d^	[Ab]	[OVA]	IC_50_	[Ab]	[OVA]	IC_50_	[Ab]	[OVA]	IC_50_
ZE*o*#1	5	30	11.1	- ^e^	-	-	10	1000	1.6	-	-	-
ZE*o*#2	5	300	12.6	-	-	-	30	1000	2.2	-	-	-
ZE*p*#1	5	100	1.0	5	30	7.1	30	1000	2.3	30	100	3.0
ZE*p*#2	5	100	1.3	5	100	3.8	10	1000	3.9	10	100	5.9

^a^ A_max_ values were between 0.5 and 1.5. ^b^ Dilution factor ×10^−3^. ^c^ Bioconjugate concentrations are in ng/mL. ^d^ Values are in nM units. ^e^ No signal was observed.

## Data Availability

Not applicable.

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
