# Peer review of "Alternative Hapten Design for Zearalenone Immunoreagent Generation"

_toxins, 2022, doi:10.3390/toxins14030185_

Round 1

Reviewer 1 Report

This manuscript reported the design, synthesis and application of zearalenone haptens. Novel haptens have longer linkers and with alternative tethering sites, which could produce polyclonal antibodies with similar sensitivity. However, the novelty of this work should be further discussed and some comparison of cross-reactivity and IC50 in literatures should be added. It could be acceptable after minor revision and checking.

Reviewer 2 Report

The manuscript titled “Alternative hapten design for zearalenone immunoreagent generation” has reviewed for consideration in toxins.  The methodology is up to standard. The results and discussion part needs more objective discussion with previous references.  

Introduction

Page 1-2: lines 26-51: The paragraphs have no references, and so much work has been carried on zearalenone. Please add references

Iqbal, S.Z*., Asi, M.R., Zia, K.M., Jinap, S. & Malik, N. (2016). A limited survey of aflatoxins and zearalenone in feed and feed ingredients from Pakistan. Journal of Food Protection, 79 (10); 1798-1801; Iqbal, S.Z*., Nisar, S., Asi, M.R. & Jinap, S. (2014). Natural incidence of aflatoxins, ochratoxin A and zearalenone in chicken meat and eggs. Food Control, 43c, 98-103; Iqbal, S.Z*.,

So please add appropriate references.

Reviewer 3 Report

Dear authors. The work you wrote is very valuable. I have little suggestions for work. In the submitted manuscript, I marked the work of errors in English and the suggestion for a sentence that is too complicated and unclear. 

Refererencej number  17 is incorrectly cited. The correct quote is: Ropejko, K., Twarużek, M. Zearalenone and its metabolites — General overview, occurrence, and toxicity. Toxins 2021, 13, 35. 
